# Isolation of Chalcomoracin as a Potential α-Glycosidase Inhibitor from Mulberry Leaves and Its Binding Mechanism

**DOI:** 10.3390/molecules27185742

**Published:** 2022-09-06

**Authors:** Yang Liu, Xue Zhou, Dan Zhou, Yongxing Jian, Jingfu Jia, Fahuan Ge

**Affiliations:** 1School of Pharmaceutical Sciences, Sun Yat-sen University, Guangzhou 510006, China; 2Chemistry and Chemical Engineering Guangdong Laboratory, Shantou 515000, China

**Keywords:** mulberry leaf, chalcomoracin, α-glucosidase inhibitor, enzymatic kinetics, molecular docking

## Abstract

Diabetes is a chronic metabolic disease, whereas α-glucosidases are key enzymes involved in the metabolism of starch and glycogen. There is a long history of the use of mulberry leaf (the leaf of *Morus alba*) as an antidiabetic herb in China, and we found that chalcomoracin, one of the specific Diels–Alder adducts in mulberry leaf, had prominent α-glucosidase inhibitory activity and has the potential to be a substitute for current hypoglycemic drugs such as acarbose, which have severe gastrointestinal side effects. In this study, chalcomoracin was effectively isolated from mulberry leaves, and its α-glucosidase inhibition was studied via enzymatic kinetics, isothermal titration (ITC) and molecular docking. The results showed that chalcomoracin inhibited α-glucosidase through both competitive and non-competitive manners, and its inhibitory activity was stronger than that of 1-doxymycin (1-DNJ) but slightly weaker than that of acarbose. ITC analysis revealed that the combination of chalcomoracin and α-glucosidase was an entropy-driven spontaneous reaction, and the molecular docking results also verified this conclusion. During the binding process, chalcomoracin went into the “pocket” of α-glucosidase via hydrophobic interactions, and it is linked with residues Val544, Asp95, Ala93, Gly119, Arg275 and Pro287 by hydrogen bonds. This study provided a potential compound for the prevention and treatment of diabetes and a theoretical basis for the discovery of novel candidates for α-glycosidase inhibitors.

## 1. Introduction

Diabetes mellitus is a chronic and metabolic disease. Every year, about 834,000 people in China die from various complications triggered by diabetes [1,2], such as cardiovascular disease [3], ketoacidosis [4], chronic kidney disease [5] and foot ulcers [6]. These diseases have brought great harm to people’s health, family happiness and the social economy [7,8,9,10]. Clinically, diabetes could be classified into two major types, and pharmacotherapy is typically used in the usual treatment regimen in type II diabetics [11]. At present, the oral hypoglycemic agents are mainly divided into three classes based on their modes of action: ① agents that promote insulin secretion, such as sulfonamides, glinides and ddp4 inhibitors [12]; ② insulin sensitizers, such as biguanides and thiazolidinediones [13] and ③ α-glycosidase inhibitors such as acarbose, voglibose and miglitol [13,14]. Among them, α-glucosidase inhibitors have been recommended as the first-line therapy for treating diabetes, and its target α-glucosidase directly participates in the metabolic pathway of starch and glycogen. α-glycosidase inhibitors bind to the intestinal mucosal epithelial cells to competitively inhibit the activity level of α-glycosidase, delay or inhibit starch’s conversion to glucose, so as to effectively reduce the blood glucose level [15]. Compared to the other hypoglycemic agents, inhibitors of α-glycosidase are milder and less drastic fluctuations in blood glucose or hypoglycemic symptoms, and the secondary disabling reactions caused by a sulfonamide drug can be avoided [16]. However, long-term use of these α-glycosidase inhibitors may cause some side effects, such as gastrointestinal reactions, liver damage, etc. Active compounds in natural products or traditional Chinese medicines (TCMs) have attracted much attention due to their low toxicity, and at the same time have advantages in the treatment of various chronic diseases with complex pathogenesis due to their complex chemical structures, naturally occurring chirality and natural affinities [15,17,18]. Thus, finding new natural products as safer and more effective α-glycosidase inhibitors might be a good choice.

Mulberry leaf is the dry leaf of *Morus alba*, which is widely cultivated from the south to north in China. As we know, mulberry is one of the most commonly used TCMs listed in the *Chinese Pharmacopoeia*, which has been proved to have satisfactory antidiabetic effects since as early as in the 11th century in China, as recorded in the *Compendium of Materia Medica*. At present, mulberry leaf tea has been developed in various forms as a functional food in the prevention and treatment of diabetes in China [19,20,21]. Modern studies have proved that mulberry leaves are rich in alkaloids, flavonoids, polysaccharides and other functional components, which can reduce blood sugar, blood pressure, blood lipids, anti-aging, anti-oxidation and anti-nervous system diseases [22,23,24,25]. Recently, “total alkaloids from *Ramulus Mori*”, the first new approved TCM for diabetes in the past 10 years in China, was put on the market. Among its main ingredients, 1-deoxyylmycin (1-DNJ), a polyhydroxy piperidine alkaloid, has been approved to be an active substance for the treatment of diabetes type II by targeting α-glucosidase [26].

The Diels–Alder adduct is another well characterized kind of ingredient in mulberry leaves, and about 22 Diels–Alder adducts have been reported to be found in *morus* species [19,20]. One of these adducts, chalcomoracin, has been shown to have antitumor activity, and its mechanism was studied in [21,27]. Recently, we found that chalcomoracin also exhibited certain activities of α-glucosidase inhibition and a hypoglycemic effect, which might have potential to be used as a substitute for existing α-glycosidase inhibitors. To our knowledge, there is no report about the inhibition activities and interaction mechanisms between Diels–Alder adducts and α-glucosidase. The low amount and extremely complicated separating process of Diels–Alder adducts may be one of the bottlenecks in performing further research.

In recent years, several methods have been developed to study the interaction mechanisms between proteins and ligands [28,29,30,31,32], such as isothermal titration (ITC), surface particle resonance, nuclear magnetic resonance wave, ultracentrifugation analysis, the fluorescence resonance energy transfer method and so on. ITC is an efficient biophysical method to study the interaction mechanisms between two molecules by calculating the heat change [33,34]. Compared with the other methods, this method is simpler, has higher efficiency, has a wider application range and is a label-free procedure. This method is gradually being applied to study interaction mechanisms between small molecules and biological macromolecules.

In this study, we propose to establish a relatively simple procedure for target-oriented isolation of chalcomoracin from mulberry leaf with high purity, combining polyamide column chromatography and preparative chromatography, then using the high purity compound to evaluate the inhibitory activities of chalcomoracin on α-glucosidase. Furthermore, the interaction mechanisms between chalcomoracin and α-glucosidase are investigated by kinetics studies, thermodynamic measurements by ITC and molecular docking simulation studies. We try to provide a simple target-oriented isolating process for the natural products and provide a theoretical basis for the discovery of novel candidates for α-glycosidase inhibitors.

## 2. Result and Discussion

### 2.1. Structure Identification of Chalcomoracin

Chalcomoracin is a pale yellow powder; [α${]}_{D}^{25}$: +84°(0.1,DMSO); HRMS: *m*/*z* = 647.2287 (M – H)^−^(calcd.forC_39_H_36_O:647.2291); 1H-NMR(600 MHz, CDCl3): 6.87(1H,s, H-3), 7.30(1H,d, *J* = 8.4 Hz), 6.66(1H,dd, *J* = 8.4, 2.0 Hz, H-5), 6.83(1H,s, H-7), 6.53(2H,s, H-2′ and H-6′), 5.44(1H,s, H-2″),4.37(1H, t, *J* = 6.2 Hz, H-4″),3.79(1H,dd, *J* = 12.4, 6.1 Hz, H-5″), 2.29–1.96(2H,m, H-6″), *J* = 8.9 Hz, H-13″),8.00(1H,d, *J* = 9.1 Hz, H-14″), 6.21(1H,d, *J* = 1.7 Hz, H-17″), 6.02(1H,dd, *J* = 8.3, 1.9 Hz, H-19″), 6.70(1H,d, *J* = 8.4 Hz, H-20″),3.05(2H,d, *J* = 6.9 Hz, H-21″), 5.02(1H,t, *J* = 7.1 Hz, H-22″), 1.60(3H, s, H-24″), 1.50(3H, s, H-25″), 1.00(t, *J* = 7.0 Hz, 2H); 13C-NMR(151 MHz, CDCl3) δ: 155.5(C-2), 102.7(C-3), 121.0(C-3a), 121.5(C-4), 112.5(C-5), 153.9(C-6), 97.4(C-7), 155.6(C-7a), 129.4(C-1′), 103.2(C-2′), 156.9(C-3′), 112.7(C-4′), 156. 9(C-5′), 103.2(C-6′), 132.2(C-1″), 127. 6(C-2″), 33.0(C-3″), 46.8(C-4″), 33.9(C-5″), 33.9(C-6″), 23.5(C-7″), 207.5(C-8″), 114.9(C-9″), 162.5(C-10″), 114.9(C-11″), 162.0(C-12″), 106.0(C-13″), 132.0(C-14″), 122.0(C-15″), 155.2(C-16″), 102.7(C-17″), 155.2(C-18″), 107.0(C-19″), 130.5(C-20″), 21.2(C-21″), 122.0(C-22″), 130.3(C-23″), 25.4(C-24″), 17.7(C-25″); this can be found in the attached Appendix A. Compared with literature data [35], the identified compound was chalcomoracin (2,4-dihydroxy-3-(3-methylbut-2-en-1-yl)phenyl)((1′R,2′S,3′R)-2,2″,4″,6-tetrahydroxy-4-(6-hydroxybenzofuran-3-yl)-5′-methyl-1′,2′,3′,4′-tetrahydro-((1,1′:3′,1″-terphenyl)-2′-yl)methanone, a known compound, and the structure is shown in Figure 1.

The extraction and separation method adopts a polyamide chromatographic column combined with preparative chromatography and does not use a large amount of toxic organic reagents in the whole process; the extraction and separation system of alcohol and water is conducive to the recovery and utilization in large-scale production, which is also in line with the development and utilization concept of mulberry leaf, a medicinal and edible Chinese medicinal material. Compared with the current separation of this compound, this method has the advantages of simplicity and greenness.

### 2.2. Inhibitory Effect on α-Glucosidase of Chalcomoracin

In this method, PNPG was used as a substrate and decomposed by α-glucosidase under weak alkaline conditions into glucose and PNG, which could be quantified via a characteristic absorption at 405 nm. Figure 2 showed the inhibition rate of chalcomoracin on α-glucosidase, as well as the control drugs of 1-DNJ and acarbose. All their inhibition rates were positively correlated with the drug concentrations, and the values of IC50 were 18.91, 14.23 and 2.34 μM for 1-DNJ, chalcomoracin and acarbose, respectively. Although the inhibitory activity of chalcomoracin was lower than that of the positive control drug Acarbose, it was superior to that of 1-DNJ, indicating that chalcomoracin has good potential as a novel glucosidase inhibitor. The result indicated that, besides alkaloids, flavonoids were also another effective part of mulberry leaves with glycosidase inhibition. In addition, chalcomoracin is one of the secondary metabolites in mulberry leaves, which is expected to reduce the gastrointestinal and hepatotoxic side effects of the chemical drug acarbose. These results provided a new idea for the development of medicinal or health care products of mulberry leaves.

### 2.3. Inhibition Mode of Chalcomoracin on α-Glucosidase

In order to study the inhibition mode of chalcomoracin on α-glucosidase, the reaction rates of the enzyme-catalyzed hydrolysis of PNPG with chalcomoracin of different quantities were investigated. Figure 3 is a linear regression relationship between 1/(S) and 1/v using the Lineweaver–Burk double reciprocal plot. According to the specific enzymatic kinetics in Table 1, it can be seen that with the increase in chalcomoracin concentration, the Michaelis constant K_m_ of chalcomoracin gradually increased, while the maximum reaction rate V_m_ gradually decreased, indicating that chalcomoracin inhibited α-glucosidase via a combination of both competitive and non-competitive ways [36].

### 2.4. ITC Determination of Interaction between Chalcomoracin and α-Glucosidase

The premise of the therapeutic effect of a drug molecule is to have a certain physical binding with the corresponding target protein in the body. Therefore, investigating the interaction between small molecules and biomacromolecules is a key technology to find the potential drugs. ITC is an advanced technique that can quantitatively describe the subtle changes in all thermodynamic parameters in the interaction between biological macromolecules and ligands and has several unique advantages such as non-specificity, small sample dosage, high sensitivity and accuracy [31]. ITC has become the preferred method for determining the interactions between biomolecules [37].

Several α-glucosidase inhibitors have been reported to be screened out from natural products using ITC without knowing the action sites, such as CSP80 from extracts of corn silk [31] and arbortristoside-C from *Nyctanthes arbor-tristis* Linn. [32].

Figure 4a shows the change in energy compensation during the titration of chalcomoracin to α-glucosidase, where the negative peaks represent an exothermic binding reaction. The area under each injection peak indicates that the cell heater needs to compensate for the electrical power supplied to the sample cell so that the temperature difference between the reference cell and the sample cell is zero, which is equal to the heat released by this drop of sample dropping into the solution. It could be seen that the peak area of the exothermic peak decreased with the titration time, indicating that the binding sites of α-glucosidase were gradually saturated with chalcomoracin molecules. Figure 4b presents a graph of the calorie molar ratio function of chalcomoracin and α-glucosidase fitted by the origin software that comes with the instrument. Figure 4c shows the thermodynamic parameters determined by ITC, including the binding stoichiometric ratio (N), equilibrium binding constant (K_a_), enthalpy change (ΔH), entropy change (ΔS) and the Gibbs free energy (ΔG), which could be calculated by ΔG = ΔH − TΔS. Their values were as follows: N = 0.261 ± 0.112, Ka = 2.52 × 10^7^ ± 4.2 × 10^8^ L·mol^−1^, ΔH = −0.9274 ± 0.116 kcal·mol^−1^, ΔS = 30.9 kcal·mol^−1^ °C^−1^, ΔG = −7.601 kcal·mol^−1^ and −TΔS = −9.579 kcal·mol^−1^. The results indicated that the chalcomoracin molecules bound to the α-glucosidase macromolecules. Normally, the thermodynamic characteristics of a spontaneous reaction at any temperature were expressed as ΔH < 0, ΔS > 0 and ΔG < 0. Moreover, the main causes of enthalpy change (ΔH) were hydrogen bonding, van der Waals forces or covalent bonding, while the entropy change (ΔS) mostly resulted from the hydrophobic interaction or changes in the spatial structure of macromolecules. Therefore, as the absolute value of −TΔS was significantly larger than ΔH, the mutual binding between chalcomoracin and α-glucosidase was a spontaneous reaction primarily driven by entropy, and the hydrophobic interaction played a leading role in the binding.

### 2.5. Binding Prediction by Molecular Docking Simulation

To further reveal the interaction behavior between α-glucosidase and chalcomoracin, a molecular docking simulation was carried out using Schrödinger 2018.4. Figure 5 shows that chalcomoracin was surrounded by some hydrophobic chains and amino acid residues including Val548, Gly546, Val544, Tyr543, Pro542, Ala97, Lys96, Asp95, Pro94, Ala93, Asp91, Pro175, Gly123, Uet122, Ala120, Gly119, Arg275, Ala289, Gly288 and Pro287 [38]. Among them, the active amino acid residues involved in hydrogen bonding were Val544, Asp95, Ala93, Gly119, Arg275 and Pro287, while Arg275 formed a cation–π interaction with a benzene ring in the chalcomoracin molecular, and the docking result is shown in Appendix A. Therefore, it could be speculated that chalcomoracin was firmly bound in a hydrophobic pocket of α-glucosidase due to the combined action of hydrophobic interaction, hydrogen bonding and cation–π interaction. This result was consistent with that of isothermal titration.

Due to the impressive pharmacological activity of alkaloids reported, Noor Rahman et al. conducted molecular docking on 32 alkaloid molecules that might have inhibition on α-glucosidase, including acarbose [39]. The results showed that acarbose bound to six amino acid residues in α-glucosidase with hydrogen binding, including Ala93, Ile98, Gln121, Met122, Arg275 and Pro545, where two residues were the same as the sites of chalcomoracin (Ala93 and Arg275). Therefore, the hydrophobic ‘pockets’ loading acarbose and chalcomoracin in α-glucosidase might be the same or adjacent, which remains to be further experimentally verified [25,36,39,40].

## 3. Materials and Methods

### 3.1. Materials and Instruments

Mulberry leaves (the leaves of *Morus alba* L.) were collected from the suburbs of Luoding City, Guangdong Province, China in April 2021; they were identified as the leaves of the *Morus alba* L. by Professor Fahuan Ge. Polyamide for column chromatography (100–200 mesh, chemical grade) was purchased from Sinopharm Shanghai Chemical Reagent Co. Ltd (Sinopharm, Shanghai, China). Ethanol (analytical grade) was purchased from Shanghai Anneiji Chemical Co., Ltd (Anneiji, Shanghai, China). P-Nitrobenzene- α-D-glucoside (PNPG) was purchased from Sigma Aldrich (Sigma, St. Louis, MO, USA). Yeast α-Glucosidase was purchased from Beijing Solaibao Technology Co., Ltd (Solaibao, Bejing, China), and 1-Deoxynojirimycin (1-DNJ) was purchased from Shanghai Aladdin Biochemical Technology Co., Ltd (Aladdin, Shanghai, China). Acarbose (lot No. ac-ef2110223, content 99.2%) was provided by Lizhu group xinbeijiang Pharmaceutical Co., Ltd (xinbeijiang, Guangzhou, China). Sodium carbonate was purchased from Shanghai Macklin Biochemical Technology Co., Ltd (Macklin, Shanghai, China). PBS buffer (pH 7.4) was purchased from BOSTER Biological Co., Ltd (Boster, Wuhan, China). A pH meter was purchased from Guangzhou Aishike Instrument Equipment Co., Ltd (Aishike, Shanghai, China).

An Essentia LC-16 liquid chromatography system (Shimadzu, Kyoto, Japan) equipped with an ACE Excel 5 C18-AR (4.6 × 250 mm, 5 μm particle size) column, a semi-preparative HPLC (Ruibai, Guangzhou, China)with model P2100 and semi-preparative C18 column (30 × 250 mm, 10 μm), a 600 M superconducting nuclear magnetic resonance spectrometer-3 (Ascend TM 500, Bruker, Germany), a liquid chromatography–mass spectrometer (UFLC-IT-TOF-MS/MS, Shimadzu, Kyoto, Japan), a high-precision polarimeter (MCP200, Anton Paar, Germany), a multifunctional microplate reader (Bio Tek, Winooski, VT, USA) and isothermal titrator (Microcal, Northampton, MA, USA) were all used.

### 3.2. Isolation of Chalcomoracin and Structure Identification

The air-dried mulberry leaf (1 kg) was crushed (passed through a 60 mesh sieve) and extracted with 80% EtOH (2 × 10 L, 2 h, 1.5 h) at 70 °C under reflux. After removal of the solvent under reduced pressure, a crude extract was obtained. The ethanolic extract was subjected to column chromatography over polyamide and eluted with ultrapure water, 30%, 60% and 85% ethanol, successively (four column volumes each solvent), each fraction being monitored by HPLC. The eluate of 85% EtOH was evaporated in vacuo to remove solvent and obtain a crude fraction (4.52 g). The crude fraction, dissolved in a mobile phase solution, was purified using preparative HPLC (MeOH:H_2_O = 80:20, flow rate 13 mL/min, wavelength 320 nm, T_R_ = 40.8 min) (Finger S10 and obtained chalcomoracin at 84.6 mg), which was identified by 1H-NMR, C13-NMR and optical rotation and compared with literature.

### 3.3. α-Glucosidase Inhibition Assay

The α-glucosidase inhibitory effect of chalcomoracin was measured using the classical PNPG method with some modifications [41]. Concisely, chalcomoracin solution with different concentrations (200, 100, 50, 25, 12.5, 6.25, 3.125 and 1.5625 μM) was prepared, while 1-DNJ and acarbose were used as positive controls. Meanwhile, solutions of α-glucosidase (0.25 U/mL), PNPG (2.5 mM) and Na_2_CO_3_ (0.2 mM) were also prepared. The inhibitory activity assay was divided into four groups including blank (A1), control (A2), sample blank (A3) and sample test (A4), and three parallel experiments were conducted for all groups. Next, PBS buffer (pH 7.4), solutions of inhibitor (chalcomoracin, 1-DNJ or acarbose) and α-glucosidase were added into a 96-well plate with quantities in Appendix A and kept in a 37 °C water bath for 10 min. Then, PNPG solution was added and kept in a 37 °C water bath for another 30 min. Finally, the Na_2_CO_3_ solution was added, and the reaction ended. The absorbance value was then measured at 405 nm on a multifunctional microplate reader, and the inhibition rate was calculated as follows:$$\mathrm{Inhibition}\;\mathrm{rate}\;\left(\%\right)=\left[1-\frac{\left(A4-A3\right)}{\left(A2-A1\right)}\right]\times 100\%$$
where *A*1 is the absorbance of the blank group, *A*2 is the absorbance of the control group, *A*3 is the absorbance of the sample blank group and *A*4 is the absorbance of the sample test group. Statistical analysis was performed with the GraphPad Prism 9.0 (Harvey Motulsky, Los Angeles, CA, USA) software to calculate the IC50 values.

### 3.4. Interaction Kinetics of α-Glucosidase Inhibition

PNPG (0.25, 0.5, 1, 2 mM) and chalcomoracin (0, 20, 50 μM) solutions of different concentrations were prepared, as well as the α-glucosidase solution (0.2 U/mL). The reaction rate was determined by plotting the enzymatic reaction rate (1/V) against the reciprocal (1/(S)) of PNPG concentration to obtain a Lineweaver–Burk double reciprocal plot to judge the type of inhibition. The relevant parameters were obtained through the following form [42]:$$\frac{1}{V}=\frac{K_{m}}{V_{max}}\frac{1}{\left[S\right]}+\frac{1}{V_{max}}$$
where *V* is the enzyme reaction rate, $V_{max}$ is maximum enzyme reaction rate, [*S*] denotes the concentration of substrate and $K_{m}$ represents the Michaelis–Menten constant.

### 3.5. Determination of Thermodynamic Parameters

The thermodynamic changes during the binding process of chalcomoracin and α-glucosidase were determined using ITC. Firstly, all the solutions of chalcomoracin and α-glucosidase were treated with ultrasonic degassing to meet the high sensitivity of ITC. Tris–HCl buffer solution (pH 7.4) was used for both the sample cell and reference cell, and the pH difference between the cells should be less than 0.1 [43]. Then, 1 mL of a 20 μM α-glucosidase solution was added into the reference cell, while 150 μL of 200 μM chalcomoracin solution was added into the sample cell. Thirty-drop samples were continuously added into the buffer solution at an addition rate of 5 μL per drop under stirring of 300 rpm, and the reaction interval was 300 s. The experiments were carried out at 37 °C, which was the temperature of the human body. The titration results were analyzed using the Origin software’s built-in ITC to obtain the thermodynamic parameter changes during the binding reaction.

### 3.6. Molecular Docking Analysis

Molecular docking analysis was carried out to predict the binding sites between chalcomoracin and α-glucosidase molecules using the MOE 2014.09 (Molecular Operating Environment) software(Chemical Computing Group ULC, Sherbrooke,Australia), which helped to further study the inhibition mechanism. MOE is a drug discovery software platform that integrates visualization, modeling and simulations, as well as methodology development, in one package. MOE scientific applications are used by biologists, medicinal chemists and computational chemists in pharmaceutical, biotechnology and academic research. The operation is as follows: download the crystal structure file of α-glucosidase with Protein Data Bank (PDB) ID of 5 kzw from the RCSB PDB database; use the MOE 2014 software to delete all solvents, ligands, etc., except for the protein; and then use the “Loop Modeler” module of the software to refer to the complete protein sequence. Complete missing fragments of proteins in crystal structure files and check for other structural errors. Then, use the Schrödinger 2018.4 (Richard Friesner, Carle Place, NY, USA) to search for the binding pocket on the processed protein and dock the small molecule into it, both of which obtain the best results recommended by the software by default. Finally, use the PyMol software (Warren Lyford DeLano, Palo Alto, CA, USA) to draw the molecular binding mode map obtained by docking.

## 4. Conclusions

In this study, a Diels–Alder addition compound in mulberry leaves, chalcomoracin, was extracted and purified using a simple and efficient manner. It was proved that chalcomoracin had a good inhibitory effect on α-glucosidase. The inhibitory mode of chalcomoracin was studied via enzymatic kinetics from the point of molecular reaction, ITC from the point of thermodynamics and molecular docking from the point of computer simulation. Based on the above studies, it was proved that chalcomoracin in mulberry leaves was expected to be developed into a α-glucosidase-targeted drug for lowering postprandial blood glucose and health food for the treatment and prevention of diabetes. However, the presented work only involved in vitro studies; further in vivo research on the hypoglycemic activity and mechanism study is expected.

## Figures and Tables

**Figure 1 molecules-27-05742-f001:**
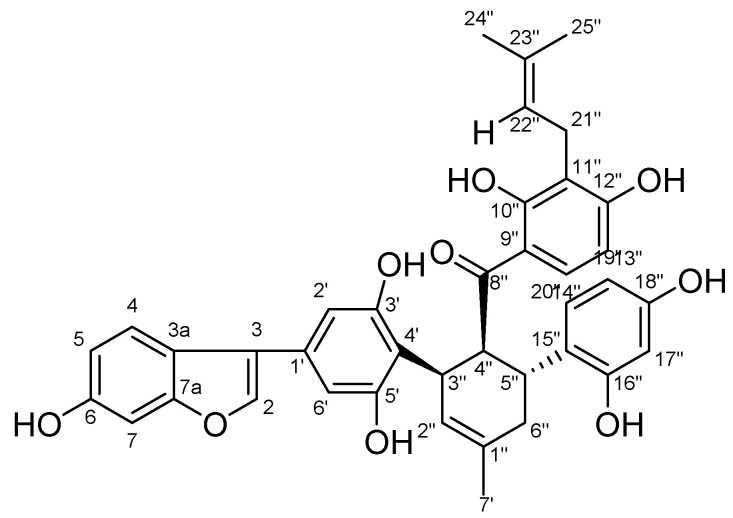
Chemical structure formula of chalcomoracin.

**Figure 2 molecules-27-05742-f002:**
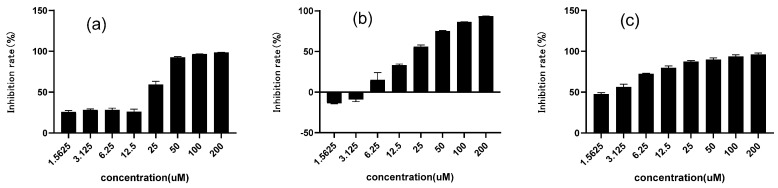
Inhibitory activity of three compounds against α-glucosidase. (**a**) Chalcomoracin; (**b**) 1-Deoxynojirimycin (DNJ); (**c**) acarbose. Data indicated as means ± SD (*n* = 3).

**Figure 3 molecules-27-05742-f003:**
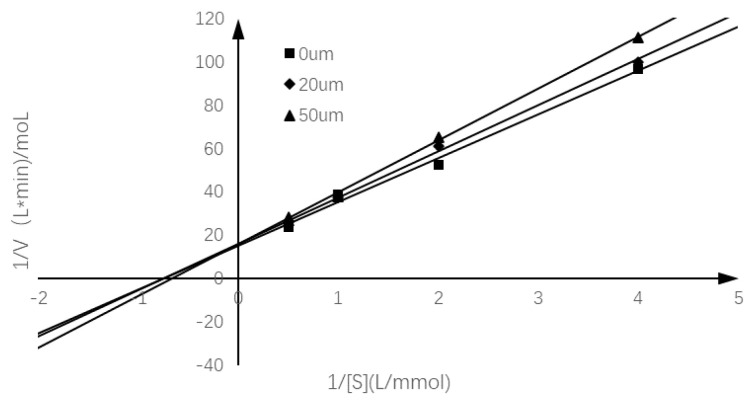
Lineweaver–Burk plot of chalcomoracin inhibition of α-glucosidase.

**Figure 4 molecules-27-05742-f004:**
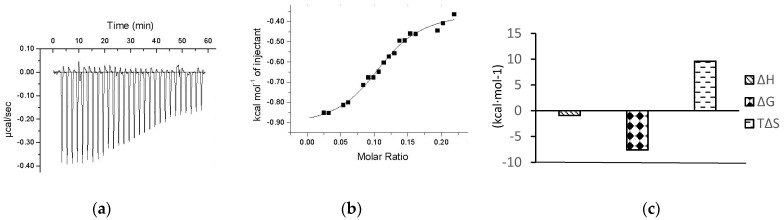
ITC thermogram for the titration of chalcomoracin to α-glucosidase. (**a**) Raw data for titration of 200 μM chalcomoracin to 20 μM α-glucosidase at 37 °C, showing the calorimetric response when chalcomoracin is continuously injected into the sample cell. (**b**) Chalcomoracin titration α-glucosidase fitting molar ratio function, enthalpy per mole of CSP80 injected versus the α-glucosidase molar ratio. The solid line represents the best non-linear least-squares fit of the independent binding sites model. (**c**) Chalcomoracin titration thermodynamic parameters of α-glucosidase.

**Figure 5 molecules-27-05742-f005:**
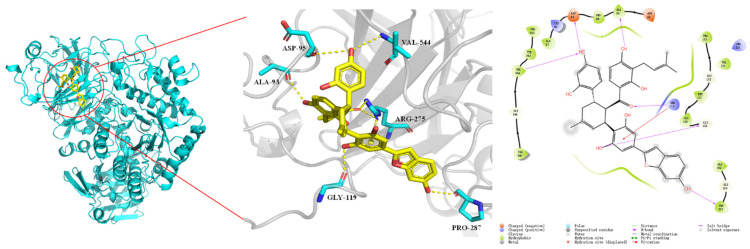
Molecular docking results of chalcomoracin and α-glucosidase.

**Table 1 molecules-27-05742-t001:** Enzymatic kinetic parameters of chalcomoracin on α-glucosidase.

Chalcomoracin Concentration	Lineweaver–Burk Equation	R^2^	−1/K_m_	K_m_	1/V_max_	V_max_
0 μM	y = 20.192x + 15.231	0.992	−0.754	1.326	15.231	0.066
20 μM	y = 21.322x + 15.901	0.9964	−0.746	1.341	15.901	0.063
50 μM	y = 23.896x + 15.914	0.9985	−0.666	1.502	15.914	0.063

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
