# Peer review of "Isolation of Chalcomoracin as a Potential α-Glycosidase Inhibitor from Mulberry Leaves and Its Binding Mechanism"

_molecules, 2022, doi:10.3390/molecules27185742_

Round 1

Reviewer 1 Report

Review: molecules-1838875

In this work, the authors have reported a set of technical methods for purifying the Diels-Alder adducts in the mulberry leaves. Additionally they discussed the α-glucosidase inhibition mechanism of chalcomoracin, a potential inhibitor of α-glucosidase that could be considered as a potential antidiabetic drug. This manuscript is important and can be published. However, publication of this manuscript in its present form is not recommended. It contains a number of confusing statements. Weak composition makes it additionally difficult to follow the paper. To be considered further for publication, this work will need to be more organized in support of the claims made in the paper. Some specific points of concern are noted below:

1) Please discuss briefly the uniqueness of the methods used in this paper.

2) Methods section should be reworded.

a) Rewrite the methods section in consistence tense. For example, the crystal structure of α-glucosidase (PDB ID 5kzw) was downloaded from the RCSB PDB database etc.

b) Spell out MOE and briefly mention the basis of the docking procedure used in this report using MOE

c) The citation for the pdb file should be included in the revised manuscript.

4) Please clarify the retention time of the target compound in the Result and Discussion section

5) Some of these docking data (in particular the molecular interactions) can be presented in the form of a table or may be included in the supplementary material.

Minor comments:

1) abstract; line 14,

“..inhibitory activity, which endowed it potential to be a substitute…” should be “…inhibitory activity, and has potential to be a substitute for current..”

2) abstract; line 15,

Remove or correct: “..and green..”

3) abstract; line 18,

“…both a competitive and a non-competitive manner..” should be “both competitive and non-competitive manners..”

4) The last sentence of the abstract should be reworded, preferably in shorter sentences

5) In the Introduction sect, pg.2; line 55-58 needs to be rearranged, reworded, preferably in shorter sentences. This applies to other sentences throughout the paper.

6) In the introduction sect,

a) pg.2; line 90 “..advantages of no label..” should be “..advantages of label-free measurement.”

b) pg.3 line 114, please put and before 85%ethaol.

7) Correct the typo on pg.4; line 156

Author Response

Response to Reviewer 1 Comments

Major and general points:

  1. Please discuss briefly the uniqueness of the methods used in this paper.

Response: Thank you for your suggestion. We are very sorry for not being able to articulate the uniqueness of methods clearly. We have rewritten introduction section carefully, hoping to clarify the uniqueness of the methods used in this paper clearly and in an appropriate way. In brief, the methods used in this paper has two highlights. Firstly, one of Diels-Alder adduct was isolated from mulberry leaves by a relatively simple method. Secondly, the inhibition mechanism between chalcomoracin and α-glucosidase has been comprehensively studied from the molecular level, kinetic level, thermodynamic point of view and molecular simulation technology. In general, this paper may provide a simple target-oriented isolating process for the natural products and provide a theoretical basis for discovery of novel candidates for α-glycosidase inhibitors.

  1. Methods section should be reworded. a) Rewrite the methods section in consistence tense. For example, the crystal structure of α-glucosidase (PDB ID 5kzw) was downloaded from the RCSB PDB database etc. b) Spell out MOE and briefly mention the basis of the docking procedure used in this report using MOE; c) The citation for the pdb file should be included in the revised manuscript.

Response: Thank you for your careful reading and reviewing of this manuscript. We feel very sorry for the inappropriate way of expression. We have re-written methods section and modify the language as possible as we can. In addition, this manuscript has been polished by MDPI editing service. All changes made to the text are highlighted in red color. Specifically, some recommendations already indicated have been modified as follows:

  1. a) We have checked the tense and modified, mainly using the past tenses for consistency.
  2. b) We have re-spelled MOE (Molecular Operating Environment) in the article and briefly touches on the basics of MOE application as follows: “MOE is a drug discovery software platform that integrates visualization, modeling and simulations, as well as methodology development, in one package. MOE scientific applications are used by biologists, medicinal chemists and computational chemists in pharmaceutical, biotechnology and academic research.”
  3. c) Thank you for your suggestion and the pdb file has been referenced in the text (Line 296, Ref 41).

  1. Please clarify the retention time of the target compound in the Result and Discussion section

Response: Thanks a lot for your suggestions. The retention time of chalcomoracin in analytical and preparative chromatography were 39.2 min and 40.8 min, respectively. We have added the preparative chromatography of last purify process and analytical HPLC chromatography in the Supporting information (see Fig.S1).

  1. Some of these docking data (in particular the molecular interactions) can be presented in the form of a table or may be included in the supplementary material.

Response: Thank you for your suggestion. The molecular interactions between chalcomoracin and α-glycosidase, according to the molecular docking results, is shown in the Supporting information (see Table S2)

Minor points:

  1. abstract; line 14, “..inhibitory activity, which endowed it potential to be a substitute…” should be “…inhibitory activity, and has potential to be a substitute for current..”
  2. abstract; line 15, Remove or correct: “..and green..”
  3. abstract; line 18, “…both a competitive and a non-competitive manner..” should be “both competitive and non-competitive manners..”
  4. The last sentence of the abstract should be reworded, preferably in shorter sentences
  5. In the Introduction sect, pg.2; line 55-58 needs to be rearranged, reworded, preferably in shorter sentences. This applies to other sentences throughout the paper.
  1. In the introduction sect, a) pg.2; line 90 “..advantages of no label..” should be “..advantages of label-free measurement.” b) pg.3 line 114, please put and before 85%ethaol.
  2. Correct the typo on pg.4; line 156

Response: Thank you for your careful reading. We have checked and modified in the text and highlighted in red color according to the referee’s suggestion.

Reviewer 2 Report

Fahuan Ge and coworkers reported on a new method to extract and purify Diels-Alder adducts from Mulberry leaves, a medicinal plant widely employed in the traditional Chinese medicine. In particular, they focused on one of these Diels-Alder adducts, called Chalcomoracin, even though is not so clear from the manuscript why/how they are able to isolate Chalcomoracin preferentially than other Diels-Alder adducts.

In addition, they assayed the inhibitory activity of Chalcomoracin (known as a potent anticancer agent) towards alpha-glucosidase to demonstrate its potential use as antidiabetic agent, finding an IC50 value of 14.23 μM and a mixed-type inhibition. They also confirm the interaction between Chalcomoracin and alpha-glucosidase via ITC, although it is not clear what additional information the ITC can provide compared to that derived from the kinetic characterization provided by the enzyme assay: is it the role played by hydrophobic interactions? Maybe the authors could specify why they decided to also exploit this technique, since it is not so diffuse for the characterization of glycosidase inhibitors. Finally, they also reported a docking simulation that suggests the interaction of Chalcomoracin with a hydrophobic pocket of alpha-glucosidase, even if is not clear if is the active site or an allosteric site.

Although the topic of the research is interesting, the manuscript requires an extensive revision of the English language because some parts are difficult to understand (e.g.: paragraph 3.5, lines 305-309; paragraph 3.4, lines 266-268, Material and Methods section) and, more importantly, it lacks scientific soundness in presenting the obtained results and in the description of the experiments (Material and Methods section). In particular, the structure of Chalcomoracin depicted in Figure 1 is not corrected (according to other reports, such as Ref 26 of the manuscript): indeed, the structure reported by authors has a stereogenic center in the cyclohexene ring (the C atom substituted with a methyl group), whose absolute configuration was not assigned. In this case the molecule would exist as a diastereomeric mixture. The authors are recommended to carefully check the structure, report the correct IUPAC name, with the R or S descriptors for the stereogenic centers present in the molecule. Addition of numbering in the depicted structure could also help the readers in the assignment of the 1H and 13C NMR signals reported. They also stated that the characterization is in agreement with the literature, but they do not mention the article(s) to which they refer: please add this information.

For the biological evaluation part, it is suggested to specify the alpha-glucosidase enzyme employed: it is supposed to be a commercial one (not human), but it would be better specifying this aspect, as well as the source (rice?).

Since the authors strongly stress on the development of an isolation method of Chalcomoracin and the use of different techniques, they are recommended to be more precise and rigorous in reporting the instruments employed for the measures and the characterization: usually a brief paragraph with general information (including trademark and model of the instruments employed) is reported at the begin of the Material and Methods. In alternative, the instrument could be mentioned in the specific paragraph where they are employed (e.g.: NMR spectrometer, MS-spectrometer and polarimeter in paragraph 2.2; the spectrophotometer employed for the enzymatic assay in paragraph 2.3 etc..).

For all these reasons, major revisions are required to the manuscript before considering its acceptance to Molecules.

In addition, some minor points must be addressed:

- The abstract is too long: please focus on the more important and innovative aspects of the present research.

- Abstract, line 8, replaced “generative” with “degenerative” (is this the sense of the sentence?).

- Line 55: the meaning of the sentence is not clear. Do the author mean “Taking advantage of….a large number…”?

- Line 70: please replace “Among its main ingredients, 1-deoxyylmycin (1-DNJ), one of the polyhydroxy piperidine alkaloids”, with “Among its main ingredients, 1-deoxyylmycin (1-DNJ), a polyhydroxy piperidine alkaloid”

- Chalcomoracin is sometimes written the first capital letter and sometimes not (chalcomoracin): please uniform the writing of the compound along the text (always write the same).

- Line 85: Rephrase the sentence because it is not correct. Do the authors mean: “The kinetic interactions of Chalcomoracin with alpha-glucosidase enzyme were studied through etc…” What does “of similar enzymatic reaction” mean?

- Line 91: The aim of the study claimed in this sentence is not consistent with the Result and Discussion, since the whole manuscript is focused on Chalcomoracin and other Diels-Alder adducts are never mentioned in the text. Please, reformulate the aim of the manuscript.

- Line 99: Please rephrase the sentence “they were identified as the leaves of the Morus alba L. that is belong Moraceae Gaudich” with “they were identified as the leaves of the Morus alba, that belong to Moraceae Gaudich”

- Lines 100-107: please replace “was from” with “was purchased from”

- Line 103: specify the source of alpha-glucosidase enzyme purchased from Beijing Solaibao Technology Co.

- Line 178: the sentence “HPLC analysis showed that the target compound in the 85% fraction of the eluent concentration had a higher response value, and the interference of other substances was minimal” doesn’t make sense if you haven’t previously read the 2.2 paragraph. HPLC analysis of what? Each sentence of the manuscript has to be self-consistent.

- Line 193: Are the authors sure that the 13C spectrum was recorded at 600 MHz? Maybe 600 MHz is the frequency for the proton, the one for carbon should be 150 MHz. Please check this data.

- Line 215: IC50 values are in general reported as mM, μM or nM, not as mol/L, μmol/L, nmol/L etc…

- Line 240: the meaning of this sentence is not clear, please rephrase it (Which is the subject of the sentence? Chalcomoracin? The enzyme?)

Author Response

Response to Reviewer 2 Comments

  1. Fahuan Ge and coworkers reported on a new method to extract and purify Diels-Alder adducts from Mulberry leaves, a medicinal plant widely employed in the traditional Chinese medicine. In particular, they focused on one of these Diels-Alder adducts, called Chalcomoracin, even though is not so clear from the manuscript why/how they are able to isolate Chalcomoracin preferentially than other Diels-Alder adducts.

Response: In the study of mulberry leaves, we found that after extracted by 80% EtOH and enriched by polyamide chromatography, the target compound was separated completely in analytical HPLC chromatography. Then, it was easily purified by preparative liquid chromatography and identified as Chalcomoracin.

Fig. S1 Liquid phase diagram during chalcomoracin preparation.

  1. In addition, they assayed the inhibitory activity of Chalcomoracin (known as a potent anticancer agent) towards alpha-glucosidase to demonstrate its potential use as antidiabetic agent, finding an IC50 value of 14.23 μM and a mixed-type inhibition. They also confirm the interaction between Chalcomoracin and alpha-glucosidase via ITC, although it is not clear what additional information the ITC can provide compared to that derived from the kinetic characterization provided by the enzyme assay: is it the role played by hydrophobic interactions? Maybe the authors could specify why they decided to also exploit this technique, since it is not so diffuse for the characterization of glycosidase inhibitors. Finally, they also reported a docking simulation that suggests the interaction of Chalcomoracin with a hydrophobic pocket of alpha-glucosidase, even if is not clear if is the active site or an allosteric site.

Response: Thanks for your careful checks. Perhaps what you want to confirm with us is the relationship between several experiments investigating the interaction of Chalcomoracin and α-Glycosidase and what each experiment was able to achieve. I will give a brief description of this as follows: 1) First, we proved that chalcomoracin has a good inhibitory activity on α-Glucosidase through α-Glucosidase inhibition assay; 2) Second, we demonstrated that chalcomoracin inhibited α-glucosidase via a combination of both competitive and non-competitive ways through kinetics of enzyme-catalyzed reactions, this means that it binds to both the active site and the allosteric site of the enzyme;3)Third, we adopted the ITC technology based on the principle that the combination of any substance will produce a change in energy, and the results show that the mutual binding between chalcomoracin and α-glucosidase was a spontaneous reaction primarily driven by entropy, and the hydrophobic interaction played a leading role in the binding;4)Finally, the molecular docking part showed two pieces of information: one, according to Ref. chalcomoracin indeed binds the amino acid residues of the active site of α-glucosidase; the other is that chalcomoracin is surrounded by the hydrophobic pocket of α-glucosidase.

  1. Although the topic of the research is interesting, the manuscript requires an extensive revision of the English language because some parts are difficult to understand (e.g.: paragraph 3.5, lines 305-309; paragraph 3.4, lines 266-268, Material and Methods section) and, more importantly, it lacks scientific soundness in presenting the obtained results and in the description of the experiments (Material and Methods section). In particular, the structure of Chalcomoracin depicted in Figure 1 is not corrected (according to other reports, such as Ref 26 of the manuscript): indeed, the structure reported by authors has a stereogenic center in the cyclohexene ring (the C atom substituted with a methyl group), whose absolute configuration was not assigned. In this case the molecule would exist as a diastereomeric mixture. The authors are recommended to carefully check the structure, report the correct IUPAC name, with the R or S descriptors for the stereogenic centers present in the molecule. Addition of numbering in the depicted structure could also help the readers in the assignment of the 1H and 13C NMR signals reported. They also stated that the characterization is in agreement with the literature, but they do not mention the article(s) to which they refer: please add this information.

Response: Thank you for your careful reading and reviewing of this manuscript. We feel very sorry for our carelessness and unprofessional descriptions in this manuscript. We have corrected and modified this manuscript as possible as we can. Furthermore, this manuscript has been polished by MDPI editing service, and we hope the revised version will meet your requirements.

The compound in this study can be confirmed as chalcomoracin, the IUPAC name of which is (2,4-dihydroxy-3-(3-methylbut-2-en-1-yl)phenyl)((1'R,2'S,3'Raa)-2,2'',4'',6-tetrahydroxy-4-(6-hydroxybenzofuran-3-yl)-5'-methyl-1',2',3',4'-tetrahydro-[1,1':3',1''-terphenyl]-2'-yl)methanone, by appearance, 1H-NMR, 13C-NMR, HRMS and optical rotation. Firstly, we checked these data carefully and they were in good agreements with the literature (Ref 38). So, we can identify the structure as the R isomer and revised the structure by adding number and stereochemical structure. Finally, to validate the results, we added the 1H and 13C NMR spectra in Supporting information (See Fig.S2).

  1. For the biological evaluation part, it is suggested to specify the alpha-glucosidase enzyme employed: it is supposed to be a commercial one (not human), but it would be better specifying this aspect, as well as the source (rice?).

Response: For biological evaluation, the source of the enzyme is very important information. The enzyme used in this study was yeast α- glucosidase, which was purchased from Beijing Solaibao Technology Co., Ltd. It has been added to the original text.

  1. Since the authors strongly stress on the development of an isolation method of Chalcomoracin and the use of different techniques, they are recommended to be more precise and rigorous in reporting the instruments employed for the measures and the characterization: usually a brief paragraph with general information (including trademark and model of the instruments employed) is reported at the begin of the Material and Methods. In alternative, the instrument could be mentioned in the specific paragraph where they are employed (e.g.: NMR spectrometer, MS-spectrometer and polarimeter in paragraph 2.2; the spectrophotometer employed for the enzymatic assay in paragraph 2.3 etc..).

Response: We feel very sorry for the inappropriate way of expression. We have re-written parts of “2.1. Materials and Instruments” and “2.2. Isolation of chalcomoracin and structure identification” in methods section. We hope that we could clarify the experimental process clearly and in an appropriate way. All the instruments used in this study were still listed in “2.1. Materials and Instruments” because we thought it will make the isolation and identification process more concisely to read.

  1. For all these reasons, major revisions are required to the manuscript before considering its acceptance to Molecules.

Response: Thank you very much for having reviewed our manuscript and giving us an opportunity to revise our manuscript. We have made careful modifications on the original manuscript. Furthermore, this manuscript has been polished by MDPI editing service, and we hope the revised version will meet the requirements of Molecules.

  1. In addition, some minor points must be addressed:

1)- The abstract is too long: please focus on the more important and innovative aspects of the present research.

2)- Abstract, line 8, replaced “generative” with “degenerative” (is this the sense of the sentence?).

3)- Line 55: the meaning of the sentence is not clear. Do the author mean “Taking advantage of….a large number…”?

4)- Line 70: please replace “Among its main ingredients, 1-deoxyylmycin (1-DNJ), one of the polyhydroxy piperidine alkaloids”, with “Among its main ingredients, 1-deoxyylmycin (1-DNJ), a polyhydroxy piperidine alkaloid”

5)- Chalcomoracin is sometimes written the first capital letter and sometimes not (chalcomoracin): please uniform the writing of the compound along the text (always write the same).

6)- Line 85: Rephrase the sentence because it is not correct. Do the authors mean: “The kinetic interactions of Chalcomoracin with alpha-glucosidase enzyme were studied through etc…” What does “of similar enzymatic reaction” mean?

7)- Line 91: The aim of the study claimed in this sentence is not consistent with the Result and Discussion, since the whole manuscript is focused on Chalcomoracin and other Diels-Alder adducts are never mentioned in the text. Please, reformulate the aim of the manuscript.

8)- Line 99: Please rephrase the sentence “they were identified as the leaves of the Morus alba L. that is belong Moraceae Gaudich” with “they were identified as the leaves of the Morus alba, that belong to Moraceae Gaudich”

9)- Lines 100-107: please replace “was from” with “was purchased from”

10)- Line 103: specify the source of alpha-glucosidase enzyme purchased from Beijing Solaibao Technology Co.

11)- Line 178: the sentence “HPLC analysis showed that the target compound in the 85% fraction of the eluent concentration had a higher response value, and the interference of other substances was minimal” doesn’t make sense if you haven’t previously read the 2.2 paragraph. HPLC analysis of what? Each sentence of the manuscript has to be self-consistent.

12)- Line 193: Are the authors sure that the 13C spectrum was recorded at 600 MHz? Maybe 600 MHz is the frequency for the proton, the one for carbon should be 150 MHz. Please check this data.

13)- Line 215: IC50 values are in general reported as mM, μM or nM, not as mol/L, μmol/L, nmol/L etc…

14)- Line 240: the meaning of this sentence is not clear, please rephrase it (Which is the subject of the sentence? Chalcomoracin? The enzyme?)

Response: Thanks for your careful checks. We feel very sorry for the inappropriate way of expression. Based on these comments and suggestions, we have made careful modifications on the original manuscript. Furthermore, this manuscript has been polished by MDPI editing service (English editing ID: english-48999). We have tried our best to revise our manuscript, and we hope the revised version will meet your requirements.

Reviewer 3 Report

The manuscript entitled “Isolation of Chalcomoracin as a Potential α-Glycosidase Inhibi- 2 tor from Mulberry Leaves and Its Binding Mechanism” by Liu et al., reports an extraction and characterization of Chalcomoracin from Mulberry leaves. Then then studied binding mechanism.

The characterization of compound is not proper. If the authors claim the titled compound to have the given chemical structure as in figure 1. What was the basis to determine its chirality? Were other diastereomers also observed. They have mentioned a single peak in HPLC. Where is the chromatogram? What is 85% fraction? First, they mentioned 4.52 g of yellow powder, then they say 84.6 mg of a yellow powder with a purity of 96%. If that is so what was the rest of the sample.

The writing also needs rigorous improvement. Line 8 chronic generative disease?

The manuscript should be carefully rewritten. A proper supplementary data file provided, then only it can be analyzed if the work merits of publication

Author Response

Response to Reviewer 3 Comments

  1. The characterization of compound is not proper. If the authors claim the titled compound to have the given chemical structure as in figure 1. What was the basis to determine its chirality? Were other diastereomers also observed. They have mentioned a single peak in HPLC. Where is the chromatogram? What is 85% fraction? First, they mentioned 4.52 g of yellow powder, then they say 84.6 mg of a yellow powder with a purity of 96%. If that is so what was the rest of the sample.

Response: Thank you for your careful reading and professional suggestion. We feel very sorry for the inappropriate way of expression. In the study of mulberry leaves, we found that after extracted by 80% EtOH and enriched by polyamide chromatography, the target compound was separated completely in analytical HPLC chromatography in 85% fraction (See Fig.S1.A). Then, it was easily purified by preparative liquid chromatography (see Fig.S1.B). The preparative chromatography of last purify process and analytical HPLC chromatography were supplemented in the Supporting information. In order to obtain the compound with high purity, we only collected the top fraction of peak in preparative chromatography. After removal of the solvent under reduced pressure, 84.6 mg of a yellow powder with a purity of 96% was obtained.

The compound in this study can be confirmed as chalcomoracin, the IUPAC name of which is (2,4-dihydroxy-3-(3-methylbut-2-en-1-yl)phenyl)((1'S,2'R,3'S)-2,2'',4'',6-tetrahydroxy-4-(6-hydroxybenzofuran-3-yl)-5'-methyl-1',2',3',4'-tetrahydro-[1,1':3',1''-terphenyl]-2'-yl)methanone, by appearance, 1H-NMR, 13C-NMR, HRMS and optical rotation. Firstly, we checked these data carefully and they were in good agreements with the literature (Ref 38). So, we can identify the structure as the R isomer and revised the structure by adding number and stereochemical structure(See Fig.1). Finally, to validate the results, we added the 1H and 13C NMR spectra in Supporting information (See Fig.S2).

Fig. S1 Liquid phase diagram during chalcomoracin preparation.

  1. The writing also needs rigorous improvement. Line 8 chronic generative disease?

Response: Thank you very much for having reviewed our manuscript and giving us an opportunity to revise our manuscript. We feel very sorry for our carelessness and unprofessional descriptions in this manuscript. We have made careful modifications on the original manuscript as possible as we can. Furthermore, this manuscript has been polished by MDPI editing service, and we hope the revised version will meet the requirements of Molecules.

Moreover, about the “chronic generative disease” in Line 8, we have replaced with “chronic metabolic disease”.    

  1. The manuscript should be carefully rewritten. A proper supplementary data file provided, then only it can be analyzed if the work merits of publication

Response: Thanks a lot for your suggestions. We have made careful modifications on the original manuscript as possible as we can.

We agree with that several supplementary data are necessary to clarify the results more clearly. We have added some diagram in Supporting information file.

In this study, we propose to establish a relatively simple procedure for target-oriented isolating chalcomoracin from mulberry leaf with high purities, combining polyamide column chromatography and preparative chromatography. Then using the high purity compound evaluate the inhibitory activities of chalcomoracin on α-glucosidase. Besides, the interaction mechanisms between chalcomoracin and α-glucosidase is investigated by kinetics studies, thermodynamic measurements by ITC, and molecular docking simulation studies. We try to provide a simple target-oriented isolating process for the natural products and provide a theoretical basis for discovery of novel candidates for α-glycosidase inhibitors.

Round 2

Reviewer 3 Report

The authors have worked on the comments and have revised the manuscript in much better form.